# Diabetes Mellitus Promotes Smooth Muscle Cell Proliferation in Mouse Ureteral Tissue through the P-ERK/P-JNK/VEGF/PKC Signaling Pathway

**DOI:** 10.3390/medicina57060560

**Published:** 2021-06-01

**Authors:** Taesoo Choi, Jeong-Woo Lee, Su-Kang Kim, Koo-Han Yoo

**Affiliations:** 1Department of Urology, College of Medicine, Kyung Hee University, Seoul 02447, Korea; taesoochoi85@daum.net (T.C.); jwleemed@hanmail.net (J.-W.L.); 2Department of Biomedical Laboratory Science, Catholic Kwandong University, Gangneung 25601, Korea; skkim7@khu.ac.kr

**Keywords:** diabetes mellitus, ureter, urolithiasis

## Abstract

*Background and objectives:* The aim of our study was to evaluate the role of diabetes mellitus (DM) as a significant factor affecting spontaneous stone expulsion, as suggested by previous research. *Materials and methods:* We investigated the influence of DM on the ureter using a murine model. The mouse-model arm of this study used 20 15 -week-old mice, including 10 normal (control) mice and 10 DM mice. We measured the proximal, middle and distal ureteral smooth muscle thickness in each mouse and the differences among ureteral sections were analyzed. Mouse ureteral specimens were also analyzed via western blotting to detect relative protein expression of phosphor–extracellular signal regulated kinases (P–ERK), phosphor–C–Jun N–terminal kinase (P–JNK), vascular endothelial growth factor (VEGF), and protein kinase C (PKC), which are representative factors involved in cell regulation. *Results:* We observed significant hyperproliferation of ureteral smooth muscle in DM mice compared to normal mice, which may provoke reduced peristalsis. The ureteral smooth muscle of DM mice was significantly thicker than that of normal mice in all ureteral tissues: proximal (*p* = 0.040), mid (*p* = 0.010), and distal (*p* = 0.028). The relative protein expression of P-ERK (*p* = 0.005) and P–JNK (*p* = 0.001) was higher in the diabetic group compared to the normal group. Additionally, protein expression of VEGF (*p* = 0.002) and PKC (*p* = 0.001) were remarkably up-regulated in DM mice. *Conclusions:* Hyperproliferation of ureteral smooth muscle was observed in DM mice, but not in normal mice. The pathways mediated by P–ERK, P–JNK, VEGF, and PKC may play an important role in pathological ureteral conditions.

## 1. Introduction

Diabetes mellitus (DM) is a carbohydrate metabolism disorder caused by various etiological factors and generally involves absolute or relative insulin insufficiency, or insulin resistance, resulting in blood glucose elevation. DM is also strongly associated with various types of genitourinary system infections, such as pyelonephritis and Fournier’s gangrene, and increases the chances of stone formation, especially those made of uric acid. A recent study of 574 patients demonstrated that DM was independently associated with the presence of stones [1].

Urinary bladder dysfunction is one of the common long-term complications of DM. It may be related to bladder dysmorphology as DM is associated with increased bladder capacity, which is triggered by decreased bladder sensation. As part of compensatory changes for impaired contractility, increased cellular proliferation was observed in patients with diabetes-associated cystopathy. There is some evidence that cell cycle dysregulation may play a role in pathological states affecting the urinary tract. One study in rabbit models demonstrated that decreased apoptosis, which is associated with increased cellular proliferation, may influence the occurrence of DM cystopathy [2].

The damaging effects of glucose on the cells are ill-understood. The sorbitol pathway, non-enzymatic glycation of proteins and increased oxidative stress, can synergize through sharing the capacity to activate mitogen-activated protein kinases (MAP kinases). There are three main groups of MAP kinases—the extracellular signal regulated kinases (ERK), the c-Jun N-terminal kinases (JNK), and the p38 kinases. In a previous study, our results suggested a strong connection between underlying DM and spontaneous stone passage failure [3]. We hypothesized that high glucose concentration in blood and urine influences ureters by altering signaling pathways, including PKC-Raf-ERK and PKC-JNK.

To the best of our knowledge, there is little available data on MAPKs in ureteral cells in diabetes. The purpose of our study was to identify the pathophysiological role of underlying DM, which is known to be a significant factor affecting spontaneous stone expulsion [3]. The pathological changes in the ureter may be a barrier for smooth urolithiasis management. Therefore, we examined MAPK expression in the ureters of diabetic mice.

## 2. Materials and Methods

### 2.1. Animal Study

Twenty 15-week-old male db/db mice (BKS.Cg-m+/+Leprdb/BomTac) were obtained from the animal supplier, Taconic Farms, for use in this study. There were 10 normal mice and 10 DM mice, weighing between 42.8 and 49.6 g. A total of 40 ureters were obtained, 20 from normal mice and 20 from DM mice. Whole environmental conditions before the experiment were identical for the two groups of mice. The animal room was maintained at 22 ± 2 °C, with 40–70% relative humidity and a 12 h light and dark cycle. All experiments were carried out according to protocols approved by the Animal Care Committee of the Animal Center at Kyung Hee University [KHUASP(SE)14-053], and in accordance with guidelines from the Korean National Health Institute of Health Animal Facility.

### 2.2. Histopathological Examination

The ureteral tissues were obtained by observing under a microscope by a urologist. Other components, such as connective tissue, were removed. Fixed ureteral tissues embedded in paraffin wax were cut into 5 μm sections and stained with hematoxylin and eosin. The sections were mounted and cover-slipped using mounting solution and then examined under a microscope. The thickness of cross-sectional segments from each ureteral section was measured. Ureteral smooth muscle area was calculated using Image Pro-Plus software (Media Cybernetic, Silver Spring, MD, USA), which made it possible to estimate the exact smooth muscle cross-sectional area. Poorly qualified images for reading were excluded.

### 2.3. Western Blotting

Ureteral tissues were processed in protein lysis buffer using a tissue homogenizer. After centrifugation at 12,000 rpm, protein was extracted and its concentration was determined using a Bradford protein assay. The proteins were transferred to membranes for western blot analysis and incubated overnight with antibodies at the following concentrations: anti-ERK (Santa Cruz, CA, USA) in a 1:200 dilution, and anti-JNK (Santa Cruz, CA, USA in a 1:200 dilution. The membranes were incubated with HRP-conjugated secondary antibody (1:5000, Pierce Chemical) for 1 h at room temperature and the membrane was developed with ECL western blotting detection reagents (GE Healthcare Biosciences, Pittsburgh, PA, USA).

### 2.4. Statistical Analyses

SPSS version 20.0 was used for statistical analyses. We employed the independent *t*-test to confirm the difference between two groups. *p*-values less than 0.05 were considered statistically significant.

## 3. Results

### 3.1. The Comparative Analysis on the Cross Sectional Area of Ureteral Smooth Muscle

The cross-sectional specimens of ureteral smooth muscle from total 20 mice, 40 ureters were obtained. Then the extent of smooth muscle component was calculated using Image Pro-Plus software. All specimens were adequate for ongoing experiment. As a result, ureteral smooth muscle in DM mice was significantly thicker than that of normal mice. There was statistically significant difference in the cross sectional area of proximal (*p* = 0.040), middle (*p* = 0.010) and distal ureter (*p* = 0.028) between the groups consistently (Figure 1).

### 3.2. The Analysis of Protein Expression by Western Blotting

Relative protein expression levels of P-JNK, P-ERK, VEGF, and PKC were analyzed by western blotting (Table 1). At first, the P-ERK expression classified into three categories according to the part of ureter that was measured. There was a consistent aspect of increase in expression levels in DM mice compared with that in normal mice (proximal, 115.08 vs. 104.54; middle, 122.99 vs. 98.83; and distal, 122.99 vs. 96.63). The average P-ERK expression level in DM mice was 120.35 ± 4.56, while that in normal mice was 100 ± 4.08. Similar trends were observed in the other pathways. High levels of P-JNK expression by part of ureter were observed depending on the presence of diabetes (proximal, 126.56 vs. 96.65; middle, 121.88 vs. 106.03; and distal, 128.57 vs. 97.32). The average value of P-JNK expression in DM mice was 125.67 ± 3.44, and that in normal mice was 100 ± 5.23. There was also the discrepancy between the two groups in VEGF expression level (proximal, 126.35 vs. 98.87; middle, 125.23 vs. 102.70; and distal, 130.18 vs. 98.42), and mean value of VEGF expression was 127.25 ± 2.60 in diabetes and 100 ± 2.35 in normal control. Further analysis of PKC expression demonstrated significant increase in DM mice (proximal, 125.81 vs. 94.93; middle, 123.96 vs. 105.53; and distal, 122.35 vs. 99.54). The mean value of PKC expression was 124.04 ± 1.73 in diabetes and 100 ± 5.31 in normal control.

All metabolic pathways were overexpressed in the ureteral smooth muscle of DM mice, and it varied from 20% to 27%. As shown in Figure 2, the differences in expression levels of P-ERK (*p* = 0.005), P-JNK (*p* = 0.001), VEGF (*p* = 0.002) and PKC (*p* = 0.001) were statistically significant higher in diabetic ureters compared with normal controls.

## 4. Discussion

DM is a systemic metabolic disease characterized by abnormal insulin production and/or insulin action over time, resulting in high blood sugar. Chronic hyperglycemia contributes to various kinds of cell damage and dysfunction, ultimately resulting in organ failure. Urological complications associated with DM can also manifest as endothelial and neural damage to the genitourinary tract. About half of all DM patients suffer from varying degrees of bladder cystopathy during their lifetime. One study reported that lower urinary tract symptoms originating from DM were found in over 80% of all DM patients [4]. Bladder dysfunction may cause symptoms such as poor bladder sensation and impaired contractility, resulting in large volumes of post-void residual urine leaking. This urine transport impairment increases the risk of urinary tract infections and urolithiasis, and renal injury, at worst. Other urological complications, such as benign prostatic hyperplasia and erectile dysfunction, have a significantly negative effect on the quality of life of diabetic patients. Life threatening conditions, such as pyelonephritis and Fournier’s gangrene, are also associated with DM. However, little is understood regarding the clinical pathway or pathophysiology of these conditions. This study experimentally confirmed that spontaneous passage of ureteral stones is difficult in diabetic patients due to thickening of the ureter. Medical expulsive therapy is one of the methods of treating small ureteral stones, but it seems difficult to apply this method to patients with DM. For patients with DM, it is more effective for clinicians to choose shock wave lithotripsy or ureterorenoscopy, which are more invasive treatments.

Urolithiasis begins as a tiny mineral deposit in the kidney. Diverse pathophysiological factors facilitate stone formation through metabolic disturbance, such as DM. The clinical correlation between urolithiasis and DM has been identified in previous studies. In 2006, a large-scale case-control study of 3561 patients evaluated the association between urolithiasis and metabolic syndrome. DM was shown to be a significant independent factor for stone formation after adjustment for age, sex, hypertension, and obesity. DM was a significant factor in the development of uric acid stones, especially [5]. Another study involved the review of 24 h urinary profiles of 1117 uric acid stone patients, with and without DM. Significantly higher uric acid and oxalate levels, as well as lower pH levels, were observed in the DM group [6].

This study experimentally concluded that DM affects the thickening of the ureter. The ureters of DM mouse and normal mouse were stained with hematoxylin and eosin to check for hyperproliferation of the ureter. Hyperproliferation of the ureter causes a decrease in the lumen of the ureter and a decrease in the mobility of the ureter. The ureter is a functional organ that transfers urine from the kidney to the bladder. In adults, the ureter is usually 25–30 cm long and 3–4 mm in diameter. Histologically, the ureter is composed of transitional epithelium and smooth muscle in the distal third. The muscle layer of the ureter plays a major role in transporting urine through involuntary peristalsis. Intact urinary flow allows for quick elimination of body waste. However, the transport of urine may be interrupted under certain pathological circumstances [7,8]. Some studies have tried to control ureteral peristalsis through medication [9]. It has been established that the contractile activity of ureteral smooth muscle is mediated by the autonomic nervous system [10]. Effective pharmaceutical modulators of ureteral peristalsis are presently unknown.

In the event of acute or chronic renal injury, the kidney activates signaling pathways, which play a part in structural and functional manifestations [11]. A few pathways, such as MAPK and mammalian target of rapamycin (mTOR), are known to be significant among the various kidney signaling networks. Some pathways, including the polyol pathway, hexosamine pathway, PKC pathway, and MAPK activation; growth factors; and cytokines have also been shown to perform important functions in diabetic nephropathy.

Similarly, a recent study showed that tight glucose control in DM patients may reduce macrovascular and microvascular complications [12]. Vascular smooth muscle cell dysfunction, which is related to high glucose levels, is a key diabetic complication [13] and may induce endothelial injury associated with increased levels of glucose transporter-1, which is involved in vascular smooth muscle cell proliferation [14]. These pathological changes ultimately result in dysfunctional hyperproliferation of vascular smooth muscle [13], and subsequent complications such as vascular stenosis and atherosclerosis.

In our study, mouse ureteral specimens were analyzed via western blotting to detect relative P-ERK, P-JNK, VEGF, and PKC protein expression. These are representative factors involved in the activation of various genes. MAPKs belong to a huge family of serine/threonine protein kinases, which are a chain of proteins that communicate signals from cell surface receptors to DNA in cell nuclei. They are involved in directing cellular responses to a diverse array of stimuli, such as mitogens, osmotic stress, heat shock, and pro-inflammatory cytokines. They also regulate cellular functions, including proliferation, differentiation, development, transformation, and survival. MAPKs activate other proteins by phosphorylating threonine and tyrosine residues through a dual-specificity MAP kinase, and serine/threonine phosphorylation activates MAP kinase by MAP kinase kinase kinases.

There are four major MAPK pathways: ERK1/2, JNK, p38, and ERK5. The ERK pathway is one of the most widely researched. It is activated by various growth factors and hormones through growth and differentiation receptors, including G-protein coupled receptors, receptor tyrosine kinases, and integrins. ERK5 is activated by both proliferative stimuli (e.g., EGF, serum, lysophosphatidic acid, and neutrophins) and stress stimuli (e.g., sorbitol, H_2_O_2_, and UV irradiation) [15]. JNK and p38 are mostly activated by cellular stress, such as exposure to UV radiation, heat shock, high osmotic stress, or cytokines. The p38 pathway shares about half of its homology with the ERK pathway. JNK, one of the stress-activated protein kinases, is a multifunctional kinase involved in gene expression in response to diverse cellular stimuli, growth factors, and cytokines. The JNK pathway has been shown to play a role in apoptosis, as well as promoting cell survival, oncogenesis, and inflammation. Its association with many pathological conditions has been recognized [16].

PKC regulates protein activity by phosphorylating hydroxyl groups in threonine and serine amino acid residues [17]. These kinases are involved in cellular proliferation and apoptosis, as well as in gene transcription and translation control. PKC is also known to play a crucial role in urethral sphincter and bladder contraction. VEGF is a signaling protein that stimulates angiogenesis [18]. It creates new vessels following vessel injury or blockage. It can be overexpressed in response to insufficient blood flow, to allow for additional oxygen supply to tissues. VEGF serum level is generally high in patients with DM [19]. Its expression in various diseases has been extensively researched in recent years. Liu et al. demonstrated that the overexpression of VEGF in glomeruli leads to hypertrophy, with the same trend identified in diabetic retinopathy [20].

The distribution of various signaling pathways within the ureter is not well understood and may vary depending on pathological state. Based on our study, these factors may be related to the hyperproliferation of ureteral smooth muscle and subsequent functional deterioration.

Our study has a few limitations. It is hard to generalize the results due to the relatively small number of animals in our experimental group. Additional research on other urinary tract tissues (e.g., kidney, bladder) will contribute to our understanding of the DM-associated urological dysfunction mechanism. To compensate for these shortcomings, a follow-up prospective, large-scale study is needed to confirm our finding that DM is a potential risk factor for the development of ureteral dysfunction. Based on our findings, evaluating patients for the presence of DM may be a useful clinical practice in predicting ureteral stone prognosis.

## 5. Conclusions

Underlying DM affects the thickening of the ureter and is known to be a cause of adverse effects on the spontaneous passage of ureteral stones. Hyperproliferation of ureteral smooth muscle was observed in DM mice, and not in normal mice. This study suggests that ERK, JNK, VEGF, and PKC signaling pathways may play a significant role in the development and progression of this pathophysiological change, resulting in ureteral dysfunction. These pathways may be potential targets for the long-term management of DM-induced urinary tract complications.

## Figures and Tables

**Figure 1 medicina-57-00560-f001:**
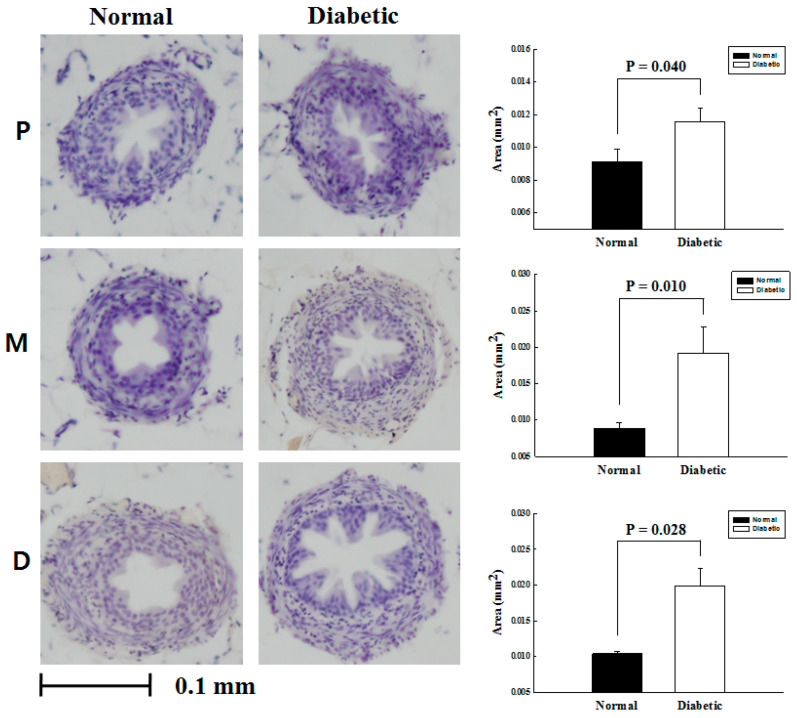
Cross-sectional images of the proximal (P), middle (M), and distal (D) ureter, with comparative graphs of normal and diabetic mouse ureteral smooth muscle thickness.

**Figure 2 medicina-57-00560-f002:**
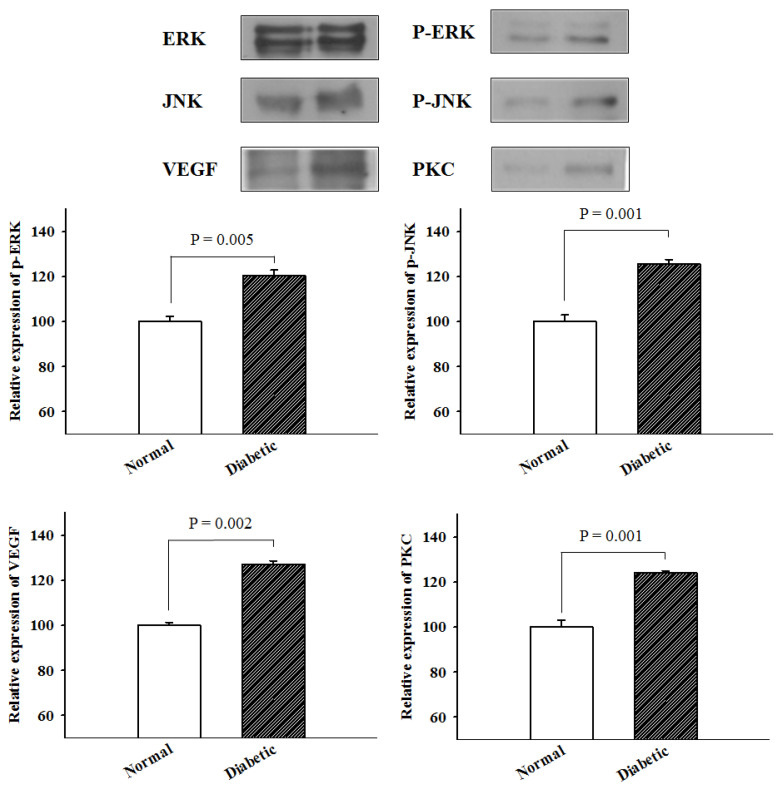
Cross-sectional images of the proximal (P), middle (M), and distal (D) ureter, with comparative graphs of normal and diabetic mouse ureteral smooth muscle thickness. P-ERK; phospho-extracellular signal regulated kinases; P-JNK: phospho-c-Jun N-terminal kinases; VEGF: vascular endothelial growth factor; PKC: protein kinase c. ERK: extracellular signal regulated kinases, JNK: c-Jun N-terminal kinases.

**Table 1 medicina-57-00560-t001:** The comparative experimental values of P-ERK, P-JNK, VEGF, and PKC between the normal and diabetic groups.

		Normal Mice			DM Mice
		Expression Level	Mean (±SD)	Expression Level	Mean (±SD)
*P-ERK*	Proximal	104.54	100 ± 4.08	115.08	120.35 ± 4.56
	Middle	98.83		122.99	
	Distal	96.63		122.99	
*P-JNK*	Proximal	96.65	100 ± 5.23	126.56	125.67 ± 3.44
	Middle	106.03		121.88	
	Distal	97.32		128.57	
*VEGF*	Proximal	98.87	100 ± 2.35	126.35	127.25 ± 2.60
	Middle	102.70		125.23	
	Distal	98.42		130.18	
*PKC*	Proximal	94.93	100 ± 5.31	125.81	124.04 ± 1.73
	Middle	105.53		123.96	
	Distal	99.54		122.35	

DM: diabetes mellitus; SD: standard deviation; P-ERK; phospho-extracellular signal regulated kinases; P-JNK: phospho-c-Jun N-terminal kinases; VEGF: vascular endothelial growth factor; PKC: protein kinase c.

## Data Availability

The data presented in this study are available on request from the corresponding author. The data are not publicly available due to privacy restrictions.

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
