# Peer review of "Diabetes Mellitus Promotes Smooth Muscle Cell Proliferation in Mouse Ureteral Tissue through the P-ERK/P-JNK/VEGF/PKC Signaling Pathway"

_medicina, 2021, doi:10.3390/medicina57060560_

Round 1

Reviewer 1 Report

This manuscript is about MAPK expression in the ureters of diabetic mice. In preclinical it might have some implications. However, it seems to be an old study (according with ethical code: KHUASP(SE)-247 14-053; approved date: February 5, 2015).

English language must to be improved.

The main questions/comments are:

  1. Why the authors what to publish so late these data?
  2. What are the future practical implications of this study? Are there?
  3. English language must to be improved.

Author Response

Reviewer 1.

All authors appreciate you reviewing manuscript. We thoroughly corrected it according to your advice.

  1. Why the authors what to publish so late these data?

- Answer: Our study is intended to know the effects of diabetes mellitus in mice. We identified diabetes mellitus as one of the causes while conducting a clinical study to find the cause of suppressing the spontaneous passage of ureteral stones. This study was planned on how diabetes mellitus affects the ureter. To confirm this experimentally, the ureter was stained with hematoxylin and eosin in mice. The thickening of the ureter was confirmed, but it was difficult to find the cause. So, western blotting was performed in a follow-up study.

This series of steps or the process of finding the cause was quite time consuming. As a result, we think it took a lot of time.

  1. What are the future practical implications of this study? Are there?

- Answer: Thank you for recommendations.

 'This study experimentally confirmed that spontaneous passage of ureteral stones is difficult in diabetic patients due to thickening of the ureter. Medical expulsive therapy is one of the methods of treating small ureteral stones, but it seems difficult to apply this method to patients with DM. For patients with DM, it is more effective for clinicians to choose shock wave lithotripsy or ureterorenoscopy, which are more invasive treatments. '

These are described in the discussion section.

  1. English language must to be improved.

- Answer: In fact, this manuscript had been corrected before. We are going to reflex your opinion, and ask for a recheck the overall grammar.

Reviewer 2 Report

Thank you for the opportunity to review a manuscript entitled “Diabetes mellitus promotes smooth muscle cell proliferation in 2 mouse ureteral tissue through the P-ERK/P-JNK/VEGF/PKC 3 signaling pathway”. The manuscript presents an animal study of mice with DM having their ureteral smooth muscle thicker than mice without DM. There are also suggested and studied certain pathophysiological pathways involved in muscle thickening. The work is very well described in all details. The topic is correctly introduced, the results are clearly presented and then appropriately discussed.

I can see few possible points to improve:

  • Abstract: In the Backgrounds, you state that DM may be a significant factor affecting spontaneous stone expulsion – So I would recommend to briefly address this postulate in the Conclusions.
  • Discussion: Aren´t there any other studies that you could compare your results with? You discuss mainly the proposed pathophysiological mechanism from your study. I recommend focusing also on your “morphological” findings in the Discussion.

Author Response

Reviewer 2.

All authors appreciate you reviewing manuscript. We thoroughly corrected it according to your advice.

  1. Abstract: In the Backgrounds, you state that DM may be a significant factor affecting spontaneous stone expulsion – So I would recommend to briefly address this postulate in the Conclusions.

- Answer: To reflect the opinion, we have added comments in conclusion, and please check the correction.

‘Underlying DM affects the thickening of the ureter and is known to be a cause of adverse effects on the spontaneous passage of ureteral stones.’

  1. Discussion: Aren´t there any other studies that you could compare your results with? You discuss mainly the proposed pathophysiological mechanism from your study. I recommend focusing also on your “morphological” findings in the Discussion.

- Answer: Thank you for your advice. To the best of our knowledge, there is not any similar study, hence we were inspired and hypothesized based on the mechanisms of diabetic vasculopathy. And we describe this on discussion section.

‘This study experimentally concluded that DM affects the thickening of the ureter. The ureters of DM mouse and normal mouse were stained with hematoxylin and eosin to check for hyperproliferation of the ureter. Hyperproliferation of the ureter causes a decrease in the lumen of the ureter and a decrease in the mobility of the ureter.’

Round 2

Reviewer 1 Report

The authors responded to my questions/comments.